# Evaluating 10 years of state-funded GP training in GP offices in Switzerland

**Kim Baumann, Fanny Lindemann, Beatrice Diallo, Zsofia Rozsnyai, Sven Streit**  *

Institute of Primary Health Care (BIHAM), University of Bern, Bern, Switzerland

* sven.streit@biham.unibe.ch

## Abstract

### Background

Switzerland lacks future general practitioners (GPs). Residents who wished to specialize as general practitioners were formerly trained solely in hospital settings. To better prepare and also attract more young doctors to become GPs, the canton of Bern (equivalent to a state) has implemented a partly state-funded vocational training program in GP practices. Our study examines the efficacy of this 10-year program, identifies factors that positively influence residents in their decision to become a GP and the distribution of new GPs in the canton of Bern, who had taken part in the traineeship.

### Methods

This cross-sectional survey among all residents, who participated in a traineeship in general practice from 2008 to 2017 in the canton of Bern asked if residents had taken a subsequent career choice as a GP and if so in which region. Residents scored the importance of their traineeship and their mentor's influence on becoming a GP. By using zip codes of work area of respondents already working as GPs and matching it with population census data, we could obtain the distribution of GPs on a per capita basis.

### Results

Out of 165 residents who participated in a traineeship, 151 (92%) completed our survey. 81% had chosen a career as a GP or were on track to become a GP. Almost half of the participants became GPs in the offices of their mentors or in the area. Our respondents emphasized the importance of their mentors' influence as well as the training program in their decision-making to become a GP. Most mentioned benefits of being a GP were broad field of medical care (37%) and a fulfilling doctor-patient relationship (34%). We could show an increase in GP practices in the canton of Bern, not only in urban but also accordingly in rural areas.

### Conclusions

Most residents continued subsequent careers as general practitioners after having completed a GP traineeship, with almost half of them in the region of their training. A vocational

**Data Availability Statement:** All relevant data are within the manuscript and its Supporting Information files.

**Funding:** This study was funded by the 'Berner Stiftung zur Förderung der Hausarztmedizin'

(HAST). HAST sponsored the work of SS and FL. The HAST foundation wishes to thank their sponsors KPT and Ärztekasse for their support. HAST, KPT and Ärztekasse had no role in study design, data collection and analysis, decision to publish, or preparation of the manuscript.

**Competing interests:** The authors have declared that no competing interests exist.

training program helped motivating young doctors to become GPs and underserved regions of the canton of Bern to gain new GPs.

## Background

General practitioners are fundamental in providing primary medical care for diverse patient populations and supporting a well-functioning health-care system. Nevertheless, Western countries, like Switzerland, are now facing a looming shortage. The canton of Bern, a geopolitical area (equivalent to a state), reflects in many ways the shortage of general practitioners faced by Western countries. In part an aging population will increase the burden on health care, with Switzerland's population aged over 65 estimated to increase by 40% in the next decade [1]. The aging population will also affect the supply of general practitioners, since over one third of all current active medical doctors will be older than 65 in the next decade [2]. The retirement of general practitioners will have a negative impact on the quality of health care, since lower mortality rates occur where there are adequate general practitioners [3]. In Bern there are approximately 800 general practitioners with a population of over one million, of which 20.5% are over 65 years of age [4].

Medical doctors in Western countries tend to practice in larger cities, leaving rural areas and remote communities underserved. This misdistribution of general practitioners has severe implications for the health care of rural communities. Underserved communities with less health care will mean poorer health outcomes [5]. Additionally, there is a negative perception of general practice in the medical field, making it one of the least popular medical specialties. This is perceived in various Western countries and influences the decision-making of residents so that a system produces doctors who disproportionally avoid general practice [6]. This misconception is underlined by common perceived challenges such as increased administrative aspects such as paperwork, phone calls and forms, generally long work hours [4, 7], and an aging population and increasing complexity of patients.

Several measures have been taken over the last decade to motivate young doctors to become GP's. In the Swiss medical system, it is not mandatory for residents to decide on a specialty before commencing training. Residents are allowed to change their decision or decide at a later point during their residency. This permits flexibility in career choices. It is also not mandatory or a regular part of the curriculum to undergo training in general practice. Therefore, the canton of Bern and the Institute of Primary Care in Bern (BIHAM) have started a partly state-funded vocational training program for residents in general practice (salaries partly paid by GP trainers, partly by the state). The program aims to provide residents clinical and practical experience in a primary care setting (GP office) for a period of 6–12 months. Additionally, primary care institutes provide programs to educate GPs in their roles as tutors and mentors [4]. GPs are able to support residents in making optimal use of professional skills and provide residents with important career-relevant information [8]. Besides personal satisfaction, there is also a professional commitment to the field of general medicine. Often if this professional relationship develops favorably, a resident may continue in the practice as a professional partner.

In this study, we aim to examine the efficacy of this 10-year program and identify factors that positively influence residents in their career decision-making process to become a GP after participation in this training program. Our study also examines the geographical distribution of GPs in the canton of Bern, who had taken part in the training program.

## Methods

### Study design

We conducted a cross-sectional study in 2018. Our study population were all residents, who participated in a traineeship in the GP training program between 2008–2017 (n = 165) in the canton of Bern, Switzerland. For training, residents were assigned to a general practitioner who provided clinical teaching and supervision in a GP practice. The length of the training for residents in months, calculated on the basis of full-time employment, averaged longer than 6 months. All GPs were required to attend a training course before being assigned a resident.

The BIHAM is the coordinating organ of the cantonal program and therefore keeps record of all participating residents. All residents, who have finished the training program, were invited via email to complete an online survey. Non-responders were contacted by telephone and invited to participate in our survey, increasing the total participation rate. Those contacted by telephone were a minority of 6%.

### Survey

Our survey was a shortened version from the questionnaire that is used by the WHM foundation (7). WHM has a long tradition to evaluate GP training program in >800 Swiss residents and is the most widely used questionnaire in Switzerland. The design of the questionnaire is regularly under audit for further improvements. Besides residents' characteristics (sex, age, years of completed traineeship prior to career choice, length of GP- training), the survey asked residents if they had taken a subsequent career choice as a GP. Using a 3- point Likert scale residents were asked to assess the importance of their GP training as well as their mentor's influence in becoming a GP. Residents who had qualified as GPs during the period the survey had been taken were asked additional questions. Besides the type of practice (solo, dual or group), they were also asked zip codes of where they were practicing. The survey contained open questions, on important factors that influenced the career choice of GPs. Two authors categorized answers by the following topics: medical care, doctor-patient relationship, work-life balance and self-employment and ranked by frequency of mentioning. Rare disagreement was sorted out by consensus.

Using zip codes provided by the responders in the survey, we could specify locations of GPs who had begun working. This novel method allowed us to draw information on the proportion of GPs starting a practice to the population of a given community By matching this data with the public population census and BAG (Swiss federal office of public health) we were able to correlate the number of GPs working in a given geographical area and therefore obtain the distribution of GPs on a per capita basis. The original survey and an English version are available (S1 Data).

### Outcome

Our primary outcome was the evaluation of determinants which influenced residents to take a subsequent career as a GP. Our secondary outcome was the evaluation of where residents who had chosen to become GPs were practicing in the canton of Bern.

### Statistical analysis

For this evaluation, we used descriptive statistics presenting proportions, means and standard deviation for parametric distributed data. We describe baseline data from the time of GP training and from the survey distributed up to 10 years after GP training. Answers to free-text open questions were analyzed in an exploratory qualitative approach coding the brief answers and

sorting them into categories. We then ordered the categories in descending order of frequency.

## Ethical approval and consent to participate

Swiss law on human research (Humanforschungsgesetz, HFG) does not require ethics committee approval to collect and analyze anonymous data. Participants gave consent to participate in our study by accessing our online survey.

## Results

We contacted 165 residents who participated in a traineeship. 89% completed the online survey. Non responders were contacted by telephone. This increased the participation rate to 92% (156).

### Baseline characteristics

From our participants 67% were female. The mean age of was 32.5 years. The majority of participants (70.9%) had between 4 to 8 years of clinical experience before commencing a residency in GP- traineeship, compared to 20% who had less than 4 years of and 19% who had more than 8 years of clinical experience. The majority of participants (70.9%) had between 6 to 8 months of traineeship in a GP office, with also half of the residents working full-time. 61.2% of participants trained in group practices compared to single practices (38.8%) (Table 1).

### Career choices

Fig 1 shows a detailed breakdown on career choices after completing the GP traineeship. 54% percent of participants continued on to a career as a general practitioner and 5% were planning to become a GP. 21% were in the process of completing their traineeships with a career in general medicine not yet ruled out.

**Table 1. Characteristics of residents at baseline of GP training program.**

| Trainee and GP training | Overall n = 165 |
| --- | --- |
| Female, n (%) | 112 (67.9) |
| Age, mean (SD) | 32.5 (3.8) |
| Years of completed traineeship as residents (%) | |
| <4 years | 33 (20.0) |
| 4–8 years | 17 (70.9) |
| >8 years | 15 (19.1) |
| Length of GP training in months (calculated on the basis of 100%), n (%) | |
| < 6 months | 10 (6.1) |
| 6–8 months | 117(70.9) |
| 9–12 months | 38 (23.0) |
| Workdays per week, n (%) | |
| 2 ½ days | 37 (22.4) |
| 3 to 4 ½ days | 45 (27.3) |
| 5 days | 83 (50.3) |
| Practice Type, n (%) | |
| Solo practice | 64 (38.8) |
| Group practice | 101 (61.2) |

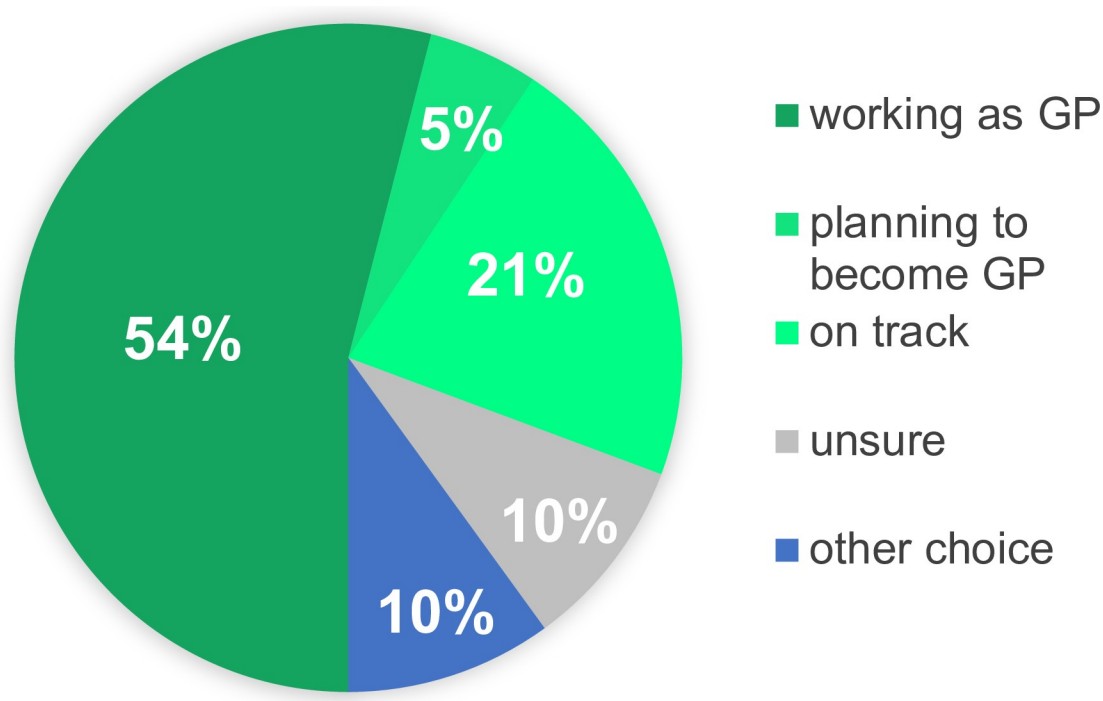

**Fig 1. Current career choice up to 10 year after participation in GP training.**

15 (10%) of trainees did not want to enter a career in General Practice. 5 had chosen anesthesia due to the possibility of working in an interdisciplinary environment of a hospital. These trainees also mentioned group practices did not provide the necessary support and felt often alone. 2 trainees had chosen psychiatry, this medical field being aligned to their interests. 3 trainees had chosen to work as general internists in a hospital setting. Their reasons for this choice were the possibility of an academic career, regulated working hours and not wanting a close long-term patient doctor relationship.

## Characteristics of new GPs and the impact of GP training on their career choice

Out of our 165 participants, 81 (54%) already worked as general practitioners by the time of the survey. When asked about the importance of their mentors influence in their decision making in becoming a GP 52 (67.5%) acknowledged that their GP mentor played an important to very important role. Only 2 (2.6%) answered with an unimportant influence. Almost half of the participants (44.9%) at the time of traineeship continued to become successors in their mentors' practice. (Table 2)

## Reasons to become GPs

From the open questions on important factors in choosing the medical profession of a GP, the most mentioned factor was the broad field of medical care (37%), closely followed by doctor-patient relationships (34%). In more detail what participants mentioned to be important were the following topics: The GP profession allows long-term care of patients, ranging from children on to adolescence and into adulthood. The doctor-patient relationship in a GP practice also allows direct contact with patients, enabling general practitioners to strengthen their

**Table 2. Characteristics of now GPs (n = 81) who completed GP training program, factors influencing decision making, workdays and successors of GP mentors' practice.**

| Characteristics, n (%) | Overall (n = 81) |
|---|---|
| **Practice Type** | |
| Solo practice | 4 (5.2) |
| Dual practice | 24 (31.2) |
| Group practice | 49 (63.4) |
| **Workdays per week** | |
| < 2 days per week | 11 (14.1) |
| 2–3 ½ days per week | 38 (48.7) |
| >4 days per week | 29 (37.2) |
| **Importance of GP training program in decision to become GP** | |
| (Very) important | 66 (85.7) |
| Indifferent | 9 (11.7) |
| (Very) unimportant | 2 (2.6) |
| **Influence of GP mentor in decision to become GP** | |
| (Very) important | 52 (67.5) |
| Indifferent | 17 (22.1) |
| (Very) unimportant | 8 (10.4) |
| **Residents who were successors of their GP mentor practice** | |
| Successor | 35 (44.9) |
| Other practice | 43 (55.1) |

medical bonds with their patients. 16% mentioned self-employment as a factor, allowing general practitioners to be more independent and flexible with their work hours.

Also, of considerable importance was work life balance (12%). Participants already working as general practitioners mentioned the possibility of balancing family and a career, when working in primary care. (Table 3)

## Location of new GP practices

In Fig 2 we can see a comparison of the number of GPs start practicing in a geographical area to general population. The majority of GPs (65%) began practicing in urban areas (large towns to cities in, for example cities like Biel or Bern). More than 20% of GPs started practices in periurban areas (for example Thun, population 43723 in 2018) and 11% of GPs started in rural areas (for example Bannwil, population, 669 in 2018) [9]. The proportion of the communities correlated on a whole to the number of GPs starting their practices in that given area.

## Discussion

This study focuses on identifying and evaluating factors that positively influenced residents in their decision-making process in becoming general practitioners. It also challenges the

**Table 3. Reasons to become GPs ranked by frequency.**

| Important factors in choosing medical profession of GP | | |
|---|---|---|
| frequency | broad category | most frequently themes |
| 37% | medical care | broad field of medical care |
| 34% | doctor patient relationship | long-term care from children to adults direct contact with patients |
| 16% | self-employment | Independency flexibility with work hours |
| 12% | work life balance | compatibility with having a family |

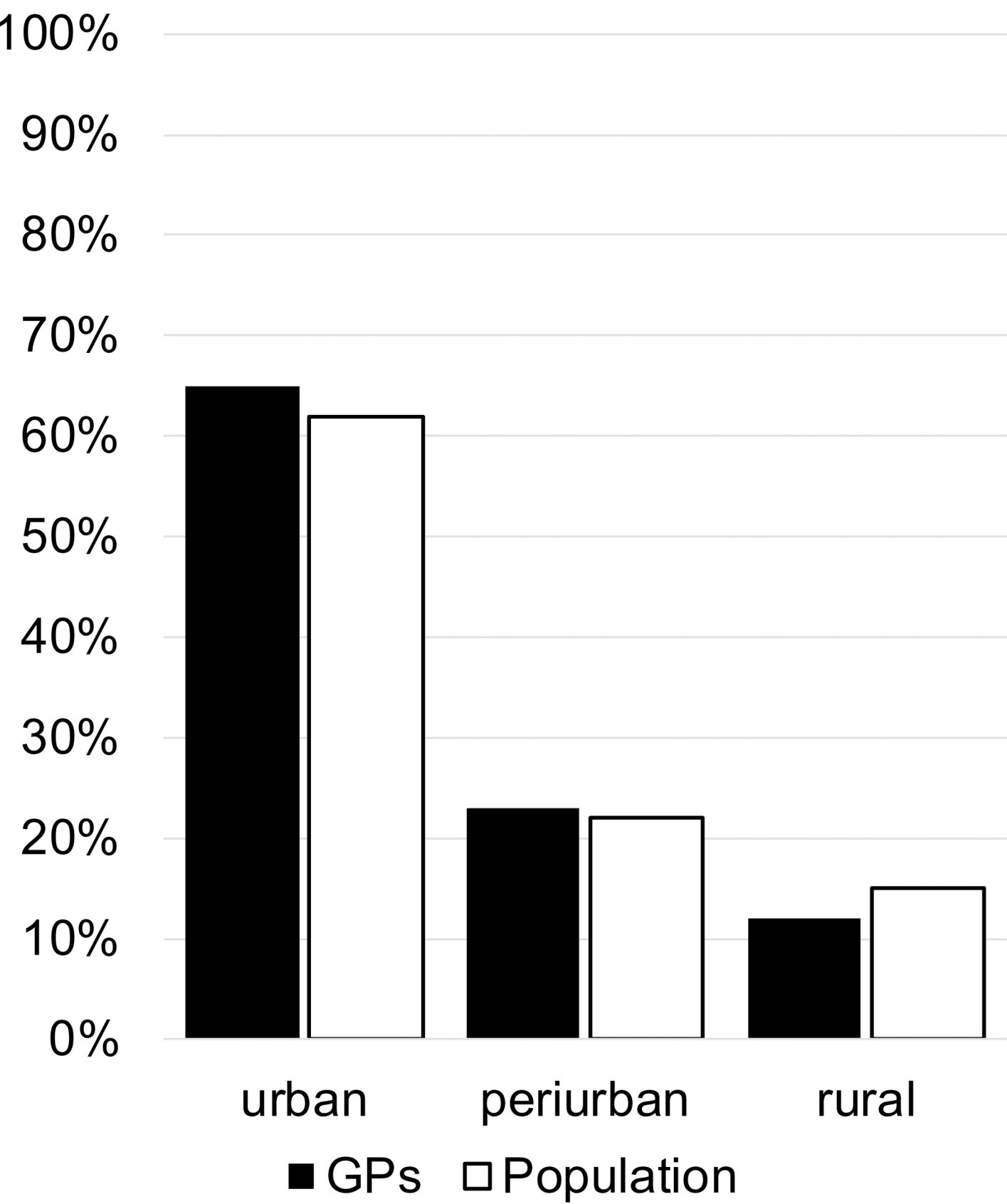

**Fig 2. Comparing proportions of where GPs start practicing (black) and where population is living (white) according to size of locations.**

assumption of an underlying chronic shortage of general practitioners in rural areas of canton Bern in Switzerland.

Firstly, these results indicate that up to 10 years after a vocational training program in a general practice a large majority of 81% of the trainees had become GPs or were on track [10]. In addition to a training program in a GP setting, mentorship also plays a significant role for success. Mentorship during traineeship is enriching and effective on a professional as well as personal level, and in the field of general medicine no exception. Other studies have also indicated that mentors can make a significant contribution to the professional development of their medical trainees [11].

Besides the possibility of part-time work (49.7%) our study also shows most residents were female (67,9%) and trained in a dual or group practice (94.6%). The positive dynamics of a group practice with flexible work hours are important factors, allowing young doctors to take on family and professional responsibilities. Group practices lead to reduced workload, better resource-sharing and a better cooperation with GPs in a practice [12, 13]. Other studies are also in line with these points and show that women consider time-related aspects and a patient orientated medical profession as important reasons for choosing general medicine [14]. But it is not only women but also men who welcome a work-life balance. In particular, a younger generation of male medical doctors highly favor predictable work hours and personal fulfilment in comparison to their predecessors [14].

Accessible and efficient health care for people living in rural areas remains an issue of ongoing concern. The canton of Bern is a geopolitical area reflecting diverse urban and rural communities speaking one or two of the four major languages (German and French). Contrary to popular belief, our study shows that GPs have been seeking employment in rural areas of the canton of Bern. There is an urban-rural disparity in physician density, however the numbers of general practitioners starting a practice in a rural region in the canton of Bern has accordingly grown to meet the demands of a smaller rural population, with smaller population growth. Swiss cities and urban communities have grown markedly over the past decade. Accordingly, a higher proportion of general practitioners have started a practice in urban communities to meet the demands of an increased population growth.

## Strengths and limitations

This study allowed an evaluation of a state-funded program up to 10 years after implementation. It was a strength that participation rate was very high (92%) lowering the risk of selection bias.

We used a novel method to document the geographical distribution of new GPs with the zip code and population-based allocation. The results also provide insight on the availability of general practitioners in rural communities by documenting distribution of GPs with zip codes allocation.

However, this study has limitations. The outcome was assessed in an anonymous survey and we were therefore unable to link baseline data with outcome data. However, we chose anonymity to reduce the risk of social desirability bias [15].

Secondly, this study examined only residents who took part in the traineeship with a general practitioner working in the canton of Bern. Nevertheless, our study results go in line with other studies in this field [10].

## Implication for research and practice

Implementing vocational training for residents with a mentor GP in a practice is not only efficient in recruiting GPs but also provides a balanced distribution of GPs also to rural areas of

canton Bern. Mentors who attend a training course can provide better support. By formalizing the process, mentoring can be more effective so that training residencies become standardized [16]. Although traineeships for residents with GPs have been implemented in other cantons of Switzerland, a thorough evaluation has not yet taken place. Implementing standardized traineeships in other cantons and providing traineeships in rural areas of Switzerland is also necessary and proved effective in recruiting young GPs. A federal run and funded traineeship as well as mentoring courses for GPs could be the next step in providing better coordination and availability for traineeships to all Swiss medical residents.

## Conclusions

A high percentage of residents continued subsequent careers as general practitioners after having completed a GP traineeship, with almost half of them in the practice of their former GP trainer. A vocational training program in general practice allows young physicians to gain prerequisite experience in this field and proved successful in motivating them to choose this career option. It also helped underserved regions in the canton of Bern to gain new GPs.

## Supporting information

**S1 Data.**
(DOCX)

**S2 Data.**
(XLSX)

## Acknowledgments

The authors thank all participants for their contribution to this study.

## Author Contributions

**Conceptualization:** Zsofia Rozsnyai, Sven Streit.

**Data curation:** Kim Baumann, Fanny Lindemann, Beatrice Diallo, Zsofia Rozsnyai, Sven Streit.

**Formal analysis:** Sven Streit.

**Funding acquisition:** Sven Streit.

**Investigation:** Fanny Lindemann, Beatrice Diallo.

**Methodology:** Kim Baumann, Zsofia Rozsnyai, Sven Streit.

**Project administration:** Zsofia Rozsnyai, Sven Streit.

**Resources:** Sven Streit.

**Software:** Sven Streit.

**Supervision:** Zsofia Rozsnyai.

**Validation:** Sven Streit.

**Visualization:** Sven Streit.

**Writing – original draft:** Kim Baumann.

**Writing – review & editing:** Kim Baumann, Fanny Lindemann, Beatrice Diallo, Zsofia Rozsnyai, Sven Streit.

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
