## [Decision Letter · Decision Letter 0]

18 May 2020

PONE-D-20-09608

Evaluating 10 years of state-funded GP training in GP offices in Switzerland

PLOS ONE

Dear Dr Streit,

Thank you for submitting your manuscript to PLOS ONE. After careful consideration, we feel that it has merit but does not fully meet PLOS ONE’s publication criteria as it currently stands. Therefore, we invite you to submit a revised version of the manuscript that addresses the points raised during the review process.

We would appreciate receiving your revised manuscript by Jul 02 2020 11:59PM. To enhance the reproducibility of your results, we recommend that if applicable you deposit your laboratory protocols in protocols.io, where a protocol can be assigned its own identifier (DOI) such that it can be cited independently in the future. For instructions see: http://journals.plos.org/plosone/s/submission-guidelines#loc-laboratory-protocols

We look forward to receiving your revised manuscript.

Kind regards,

Andrew Soundy

Academic Editor

PLOS ONE

Journal Requirements:

4. Your ethics statement must appear in the Methods section of your manuscript. If your ethics statement is written in any section besides the Methods, please move it to the Methods section and delete it from any other section. Please also ensure that your ethics statement is included in your manuscript, as the ethics section of your online submission will not be published alongside your manuscript.

Additional Editor Comments (if provided):

Can you detail steps taken in survey development if it was a new survey e.g.,

Questionnaire design and pretesting

•What past questionnaires have been designed and what questions have been posed?

•The pre-test and undertake revisions across several drafts

•Early stages of pre-test get people to comment on questions

•Use cognitive interviewing (what thoughts occurred as questions were asked) or focus groups (get people together 6-12 to talk about the questions)

•Later stages get people to complete the survey

•Other considerations

•Standardization of training material

•Debrief after each round

•Identify aspects which were important (follow up on contacts and response rate)

If the survey was taken from another study can you reference that and check for quality processes

Can you consider non-sampling error

Can you use STROBE to detail sections of the method and identify important aspects like bias, confounding variables etc.

Reviewers' comments:

Reviewer's Responses to Questions

**Comments to the Author**

1. Is the manuscript technically sound, and do the data support the conclusions?

Reviewer #1: Yes

Reviewer #2: Yes

2. Has the statistical analysis been performed appropriately and rigorously? 

Reviewer #1: Yes

Reviewer #2: Yes

3. Have the authors made all data underlying the findings in their manuscript fully available?

Reviewer #1: Yes

Reviewer #2: Yes

4. Is the manuscript presented in an intelligible fashion and written in standard English?

Reviewer #1: No

Reviewer #2: Yes

5. Review Comments to the Author

Reviewer #1: Thank you for your insight in the influences of the GP-training program on (future) GP's. I have some questions for further optimizing your paper:

- I would like to advise you to consider using a translation agency or a native speaker in order to optimize the use of language in your paper.

- In the lines 83-88 you mention the length of the training in months. Whose training duration is mentioned? As you mention in the next sentence that trainers also have to complete a training course, it is not clear to me if the length of the training is related to the trainer or the trainee.

- In line 123 you mention the use of a qualitative approach. Which qualitative approach did you use in analysing your data? (e.g. exploratory, thematic analysis, etc.)

- For consideration: would you advise to implement GP-internships for medical students in order to raise the attention for general practice already during the course of the medicine study?

Reviewer #2: This article provides an interesting review on a new GP training programme. Please see suggestions below:

Abstract line 3

May be good to explain the term “canton” when first mentioned.

Background line 53

You refer to general medicine as a specialty, rather than general practice/family medicine. In the UK general medicine is a separate specialty from general practice and it would be good to clarify if the issues you are referring to apply to all generalist fields e.g. general medicine in hospital and general practice.

Methods line 85

Need to insert comma after training: “For training residents were assigned to a general practitioner who provided clinical teaching and supervision in a GP practice.”

Methods line 88

Add in “being” between “before” and “assigned”.

“GPs were required to attend a training course before assigned a resident.”

Survey line 99

Insert apostrophe in “mentor’s”.

“importance of their GP training as well as their mentors influence in becoming a GP.”

Survey line 103

Insert full stop after “GPs.”:

“factors that influenced the career choice of GPs Two authors categorized answers by the”

Baseline characteristics line 137

Change comma to full stop: “(38,8%)”

Career choices line 144

“15 (10 %) of trainees did not perceive a career in General Practice.” Change “did not perceive” to “did not expect to enter” or “did not want to enter a career in general practice”.

Reasons to become GPs line 164

“were following topics”. Insert “were THE following topics”.

Location of new GP practices line 178

“The proportion of the communities correlated on a whole to the increase of GPs starting their practices in that given area.”

I think this should say” correlated on a whole to the number of GPs” rather than “increase of GPs”. Do you have figures for how many GPs per population there were before the study to show that the proportion of GPs working in rural areas was previously lower?

Discussion line 187

“Firstly, these results indicate that a vocational training program in a general practice increases the likelihood of a career in general practice.”

To say that it increases the likelihood you would need baseline figures, given there was not a previous training programme this is difficult to determine but perhaps looking at the number of new GPs starting in the area pre and post the training scheme?

Discussion line 197

“Group practices lead to reduced work-load, better resource-sharing and a better cooperation with GPs in a practice.”

Please add references to these points.

Implication for research and practice line 234

“mentoring can be more effective so that training residents becomes standardized”

Should this be “residencies” or “traineeships” rather than “residents”?

6. PLOS authors have the option to publish the peer review history of their article (what does this mean?). If published, this will include your full peer review and any attached files.

Reviewer #1: Yes: Linda Bonnie, PhD-student, GP-trainee

Reviewer #2: Yes: Claire Patricia Rees

---

## [Author Response · Author response to Decision Letter 0]

26 Jun 2020

Professor Sven Streit, MD, MSc, PhD

Professor in Primary Care

Head of Interprofessional Primary Care

Institute of Primary Health Care Bern (BIHAM)

University of Bern

Mittelstrasse 43

3012 Bern

Switzerland

Tel +41 31 631 58 75

Email: sven.streit@biham.unibe.ch

Revision of manuscript ID: PONE-D-20-09608

Dear Dr Soundy, dear PLOS ONE Editorial Board,

We were invited to submit a revised version of our manuscript: 

“Evaluating 10 years of state-funded GP training in GP offices in Switzerland

We would like to thank you for the opportunity to revise and resubmit a revised version of our manuscript. 

We believe the comments and suggestions made by both reviewers helped to improve our manuscript and we were happy to implement them in our manuscript. Please find a detailed response to the questions and comments raised by reviewers in our point-by-point response below.

Again, we thank you for the opportunity to strengthen our manuscript with your valuable comments. We hope that the revisions persuade you to accept our submission.

Sincerely,

Sven Streit

 

Journal Requirements:

Response: we adapted our revision accordingly. 

Response: We added our original survey and an English version as supporting information and added on line 112: 

" The original survey and an English version is available (Supporting Information 1). " 

Response: we made sure both the abstract in the online submission and the manuscript are identical. 

4. Your ethics statement must appear in the Methods section of your manuscript. If your ethics statement is written in any section besides the Methods, please move it to the Methods section and delete it from any other section. Please also ensure that your ethics statement is included in your manuscript, as the ethics section of your online submission will not be published alongside your manuscript.

Response: We moved the ethics statement as requested to the methods section. 

Editor Comments

1. Can you detail steps taken in survey development if it was a new survey e.g.,

Questionnaire design and pretesting

Response: We agree that further clarification as rightly pointed out are mandatory. We used ta shortened version of he most established national questionnaire issued by the WHM foundation (our reference 7). We believe for readers this additional information is helpful so we added on page 6: 

" Our survey was a shortened version from the questionnaire that is used by the WHM foundation (7). WHM has a long tradition to evaluate GP training program in >800 Swiss residents and is the most widely used questionnaire in Switzerland. The design of the questionnaire is regularly under audit for futher improvements. "

2. Can you consider non-sampling error

Response: There is always a risk of a non-sampling error when doing survey. But as pointed out in line 226 of our limitation section: only 8% of invited GP trainees did not answer to the survey why we believe the risk of a non-sampling error is small. 

3. Can you use STROBE to detail sections of the method and identify important aspects like bias, confounding variables etc.

Response: We provide a STROBE checklist to make sure all aspects mentioned are covered in our manuscript. 

Reviewer 1

1. Thank you for your insight in the influences of the GP-training program on (future) GP's. I have some questions for further optimizing your paper:

- I would like to advise you to consider using a translation agency or a native speaker in order to optimize the use of language in your paper.

Response: Thank you for reviewing our paper. We have taken your suggestions to improve language into consideration and have made the necessary changes.

2. In the lines 83-88 you mention the length of the training in months. Whose training duration is mentioned? As you mention in the next sentence that trainers also have to complete a training course, it is not clear to me if the length of the training is related to the trainer or the trainee.

Response: Thank you for your comment. We adapted on line 87:

" The length of the training for residents in months, calculated on the basis of full-time employment, averaged longer than 6 months. All GPs were required to attend a training course before assigned a resident."

3. In line 123 you mention the use of a qualitative approach. Which qualitative approach did you use in analysing your data? (e.g. exploratory, thematic analysis, etc.)

Response: We are glad to clarfiy: We were open to see reasons of the participants for choosing primary care and therefore included open-ended questions that we analyed in an exploratory analysis. We added this information in line 127.

4. For consideration: would you advise to implement GP-internships for medical students in order to raise the attention for general practice already during the course of the medicine study?

Response: We share the hypothesis with the Reviewer. In Switzerland, we see almost all universities having implemented GP internships but no firm evaluation to estimate the effect on career choices. 

 

Reviewer 2: 

1. This article provides an interesting review on a new GP training programme. Please see suggestions below:

Abstract line 3

May be good to explain the term “canton” when first mentioned.

Response: Thank you for your comment. On page 4 in line 39 canton is explained as being a geopolitical area equivalent to a state. We did not explain canton in the abstract, because of word count restrictions and kept the abstract concise and factual.

2. Background line 53

You refer to general medicine as a specialty, rather than general practice/family medicine. In the UK general medicine is a separate specialty from general practice and it would be good to clarify if the issues you are referring to apply to all generalist fields e.g. general medicine in hospital and general practice.

Response: In Switzerland general internal medicine is a medical speciality. After having obtained board certification for this specialty, medical doctors are allowed to work in various fields of general medicine, such as in a hospital setting or a general practice. General practice though is not a separate specialty. 

To be more clear, we adapted on page 5 in lines 52-56: “

Additionally, there is a negative perception of general practice in the medical field, making it one of the least popular medical specialties. This is perceived in various Western countries and influences the decision-making of residents so that a system produces doctors who disproportionally avoid general practice [6].

3. Methods line 85

Need to insert comma after training: “For training residents were assigned to a general practitioner who provided clinical teaching and supervision in a GP practice.”

Response: adapted. 

4. Methods line 88

Add in “being” between “before” and “assigned”.

“GPs were required to attend a training course before assigned a resident.”

Response: adapted.

5. Survey line 99

Insert apostrophe in “mentor’s”.

“importance of their GP training as well as their mentors influence in becoming a GP.”

Response: adapted.

6. Survey line 103

Insert full stop after “GPs.”:

“factors that influenced the career choice of GPs. Two authors categorized answers by the”

Response: adapted.

7. Baseline characteristics line 137

Change comma to full stop: “(38,8%)”

Response: adapted.

8. Career choices line 144

“15 (10 %) of trainees did not perceive a career in General Practice.” Change “did not perceive” to “did not expect to enter” or “did not want to enter a career in general practice”.

Response: adapted.

9. Reasons to become GPs line 164

“were following topics”. Insert “were THE following topics”.

Response: adapted.

10. Location of new GP practices line 178

“The proportion of the communities correlated on a whole to the increase of GPs starting their practices in that given area.”

I think this should say” correlated on a whole to the number of GPs” rather than “increase of GPs”. 

Response: adapted.

11. Do you have figures for how many GPs per population there were before the study to show that the proportion of GPs working in rural areas was previously lower?

Response: We agree with the Reviewer that this information would be really helpful. We find it challenging to explain to colleagues from different countries, that Switzerland has a huge problem in providing those data. E.g. in the Canton of Bern there is no up-to-date registry on active GPs which makes it impossible to have an estimate on previous numbers of GPs by geographical region. 

12. Discussion line 187

“Firstly, these results indicate that a vocational training program in a general practice increases the likelihood of a career in general practice.”

To say that it increases the likelihood you would need baseline figures, given there was not a previous training programme this is difficult to determine but perhaps looking at the number of new GPs starting in the area pre and post the training scheme?

Response: We agree with the concern of the reviewer and given our lack of data as described above, we rephrased on line 203-205: 

" Firstly, these results indicate that up to 10 years after a vocational training program in a general practice a large majority of 81% of the trainees had become GPs or were on track [10]. "

13. Discussion line 197

“Group practices lead to reduced work-load, better resource-sharing and a better cooperation with GPs in a practice.”

Please add references to these points.

Response: We added two new references (our reference 12 and 13). 

14. Implication for research and practice line 234

“mentoring can be more effective so that training residents becomes standardized”

Should this be “residencies” or “traineeships” rather than “residents”?

Response: adapted.

---

## [Decision Letter · Decision Letter 1]

14 Jul 2020

PONE-D-20-09608R1

Evaluating 10 years of state-funded GP training in GP offices in Switzerland

PLOS ONE

Dear Dr. Streit,

Thank you for submitting your manuscript to PLOS ONE. After careful consideration, we feel that it has merit but does not fully meet PLOS ONE’s publication criteria as it currently stands. Therefore, we invite you to submit a revised version of the manuscript that addresses the points raised during the review process.

Please see comments identified by the reviewers.

We look forward to receiving your revised manuscript.

Kind regards,

Andrew Soundy

Academic Editor

PLOS ONE

Reviewers' comments:

Reviewer's Responses to Questions

**Comments to the Author**

1. If the authors have adequately addressed your comments raised in a previous round of review and you feel that this manuscript is now acceptable for publication, you may indicate that here to bypass the “Comments to the Author” section, enter your conflict of interest statement in the “Confidential to Editor” section, and submit your "Accept" recommendation.

Reviewer #1: All comments have been addressed

Reviewer #2: (No Response)

2. Is the manuscript technically sound, and do the data support the conclusions?

Reviewer #1: Yes

Reviewer #2: Yes

3. Has the statistical analysis been performed appropriately and rigorously? 

Reviewer #1: Yes

Reviewer #2: Yes

4. Have the authors made all data underlying the findings in their manuscript fully available?

Reviewer #1: No

Reviewer #2: Yes

5. Is the manuscript presented in an intelligible fashion and written in standard English?

Reviewer #1: Yes

Reviewer #2: Yes

6. Review Comments to the Author

Reviewer #1: Thank you for your revision of the manuscript. You have reacted on all my concerns about the manuscript. I have one question left: Your data are available upon reasonable request, can you please indicate why your data are not fully available?

Reviewer #2: Please see a few remaining minor comments:

Abstract

Page 2 line 23 ‘importance of their mentors influence’- insert apostrophe after mentors’

Also for Page 10 line 165

Background

Page 4 line 52 and 55- in response to reviewer comments you said you were changing reference to general medicine to ‘general practice’ but it still says general medicine.

Page 11 line 205 ‘Besides the possibility of part-time work (49.7%) our study shows most residents were female (67.9%) and trained in a dual or group practice (94.6%).’ Do you mean almost half of respondents were attracted to general practice by the possibility of part-time work?

Page 12 line 217- ‘speaking one or both of the four major languages (German and French)’- should this say ‘one or two of’.

Page 12 line 222- ‘und’ should be ‘and’.

7. PLOS authors have the option to publish the peer review history of their article (what does this mean?). If published, this will include your full peer review and any attached files.

Reviewer #1: No

Reviewer #2: **Yes: **Claire P Rees

---

## [Author Response · Author response to Decision Letter 1]

14 Jul 2020

Professor Sven Streit, MD, MSc, PhD

Professor in Primary Care

Head of Interprofessional Primary Care

Institute of Primary Health Care Bern (BIHAM)

University of Bern

Mittelstrasse 43

3012 Bern

Switzerland

Tel +41 31 631 58 75

Email: sven.streit@biham.unibe.ch

Revision of manuscript ID: PONE-D-20-09608R1

Dear Dr Soundy, dear PLOS ONE Editorial Board,

We were invited to submit a 2nd revision version of our manuscript: 

“Evaluating 10 years of state-funded GP training in GP offices in Switzerland

We are glad to see that our revision satisfied the two reviewers. However, we appreciate the opportunity to address all their remaining comments. 

Sincerely,

Sven Streit

Reviewer 1

1. Thank you for your revision of the manuscript. You have reacted on all my concerns about the manuscript. I have one question left: Your data are available upon reasonable request, can you please indicate why your data are not fully available?

Response: We thank the reviewer for the positive response. The cited statement on data availability was part of our first submission. However, when we submitted the revised manuscript, we were asked to by the Editorial team to make our data publically available which we did. We would like to point to page 14 line 266 where it says: 

" All relevant data are within the manuscript and its Supporting Information files."

The file supporting information 2 contains all data. 

Reviewer 2: 

1. Please see a few remaining minor comments: Abstract Page 2 line 23 ‘importance of their mentors influence’- insert apostrophe after mentors’

Response: adapted

2. Also for Page 10 line 165. Background Page 4 line 52 and 55- in response to reviewer comments you said you were changing reference to general medicine to ‘general practice’ but it still says general medicine.

Response: our appologies, we adapted and rephrased "general practice" in line 52 and 55. 

Page 11 line 205 ‘Besides the possibility of part-time work (49.7%) our study shows most residents were female (67.9%) and trained in a dual or group practice (94.6%).’ Do you mean almost half of respondents were attracted to general practice by the possibility of part-time work?

Response: As explained in line 213-214 we know from a previous study (our reference 14) that working part-time is a factor that young doctors are attracted to general practice. 

Page 12 line 217- ‘speaking one or both of the four major languages (German and French)’- should this say ‘one or two of’.

Response: we adapted in line 219 "one or two of the four major languages". 

Page 12 line 222- ‘und’ should be ‘and’.

Response: well spotted. We again searched the complete documents for typos and adapted in line 224 "and" instead of "und".

---

## [Decision Letter · Decision Letter 2]

29 Jul 2020

Evaluating 10 years of state-funded GP training in GP offices in Switzerland

PONE-D-20-09608R2

Dear Dr. Streit,

We’re pleased to inform you that your manuscript has been judged scientifically suitable for publication and will be formally accepted for publication once it meets all outstanding technical requirements.

Kind regards,

Andrew Soundy

Academic Editor

PLOS ONE

Additional Editor Comments (optional):

Reviewers' comments:

Reviewer's Responses to Questions

**Comments to the Author**

1. If the authors have adequately addressed your comments raised in a previous round of review and you feel that this manuscript is now acceptable for publication, you may indicate that here to bypass the “Comments to the Author” section, enter your conflict of interest statement in the “Confidential to Editor” section, and submit your "Accept" recommendation.

Reviewer #1: All comments have been addressed

2. Is the manuscript technically sound, and do the data support the conclusions?

Reviewer #1: Yes

3. Has the statistical analysis been performed appropriately and rigorously? 

Reviewer #1: Yes

4. Have the authors made all data underlying the findings in their manuscript fully available?

Reviewer #1: Yes

5. Is the manuscript presented in an intelligible fashion and written in standard English?

Reviewer #1: Yes

6. Review Comments to the Author

Reviewer #1: Thank you for resubmitting your article, it is nice to have the data available next to the article. Thank you very much for all your efforts!

7. PLOS authors have the option to publish the peer review history of their article (what does this mean?). If published, this will include your full peer review and any attached files.

Reviewer #1: No

---

## [Editor Report · Acceptance letter]

5 Aug 2020

PONE-D-20-09608R2 

Evaluating 10 years of state-funded GP training in GP offices in Switzerland 

Dear Dr. Streit:

I'm pleased to inform you that your manuscript has been deemed suitable for publication in PLOS ONE. Congratulations! Your manuscript is now with our production department. 

Kind regards, 

on behalf of

Dr. Andrew Soundy 

Academic Editor

PLOS ONE